# Editors' perspectives on the peer-review process in biomedical journals: protocol for a qualitative study

Ketevan Glonti,[1,2] Darko Hren[1]

[1]School of Humanities and Social Sciences, University of Split, Split, Croatia
[2]INSERM, U1153 Epidemiology and Biostatistics Sorbonne Paris Cité Research Center (CRESS), Methods of therapeutic evaluation of chronic diseases Team (METHODS), Paris Descartes University, Paris, France

**Correspondence to**
Ketevan Glonti; kglonti@unist.hr

## ABSTRACT

**Introduction**  Despite dealing with scientific output and potentially having an impact on the quality of research published, the manuscript peer-review process itself has at times been criticised for being 'unscientific'. Research indicates that there are social and subjective dimensions of the peer-review process that contribute to this perception, including how key stakeholders—namely authors, editors and peer reviewers—communicate. In particular, it has been suggested that the expected roles and tasks of stakeholders need to be more clearly defined and communicated if the manuscript review process is to be improved. Disentangling current communication practices, and outlining the specific roles and tasks of the main actors, might be a first step towards establishing the design of interventions that counterbalance social influences on the peer-review process.  The purpose of this article is to present a methodological design for a qualitative study exploring the communication practices within the manuscript review process of biomedical journals from the journal editors' point of view.

**Methods and analysis**  Semi-structured interviews will be carried out with editors of biomedical journals between October 2017 and February 2018. A heterogeneous sample of participants representing a wide range of biomedical journals will be sought through purposive maximum variation sampling, drawing from a professional network of contacts, publishers, conference participants and snowballing.  Interviews will be thematically analysed following the method outlined by Braun and Clarke. The qualitative data analysis software NVivo V.11 will be used to aid data management and analysis.

**Ethics and dissemination**  This research project was evaluated and approved by the University of Split, Medical School Ethics Committee (2181-198-03-04-17-0029) in May 2017. Findings will be disseminated through a publication in a peer-reviewed journal and presentations during conferences.

## INTRODUCTION

Most journals in the biomedical field implement a prepublication process which primarily involves the interaction of three key stakeholders around an academic research manuscript: journal editors, peer reviewers and authors. This process, typically referred to as 'peer review', is strongly embedded in the field of biomedical

## Strengths and limitations of this study

► Qualitative analysis of interview data from a wide range of editors of biomedical journals will allow an in-depth understanding of the communication practices prevailing within biomedical journals.
► Quality assurance will be employed throughout data collection and analysis to ensure traceability and verification.
► Journal editors of a selection of biomedical journals will be interviewed; therefore, research findings cannot directly be extrapolated to all biomedical journals and other scientific fields.

publishing and in its broadest sense refers to the evaluation of manuscripts submitted for publication by researchers from the same or related areas of expertise. Thus far, there is no universal definition of 'peer review', and its specific objectives are not clearly defined.[1] Concurrently, the roles, tasks and core competencies expected of peer reviewers and editors have not been formally established and both groups operate largely without formal training.[2 3] A study that aimed to identify all tasks that journal editors expect of peer reviewers who evaluate a manuscript reporting a randomised controlled trial (RCT) found that the most important tasks in peer review, as perceived by peer reviewers evaluating RCTs, were not congruent with the tasks most often requested by journal editors in their guidelines to reviewers.[4] These differences illustrate the need to clarify the roles and tasks of peer reviewers.

The peer-review process has at times been criticised for being 'unscientific'.[5 6] Despite dealing with scientific output that potentially leads to changes in clinical practice, the process itself is not without potential biases—including prestige or association bias, gender bias, confirmation bias, conservatism, bias against interdisciplinary

research, publication bias, language bias and conflict of interest.[7 8]

In light of this criticism, there have been several attempts to improve the peer-review process and the quality of peer-reviewer reports in the biomedical field.[9] A recent systematic review evaluating the impact of interventions to improve the quality of peer review for biomedical publications[10] identified 25 strategies that have been implemented, including training interventions; use of checklists (such as Consolidated Standards of Reporting Trials[11]); addition of specific experts (ie, statistical peer reviewers); the introduction of open peer review (ie, peer reviewers informed that their identity would be revealed) or blinded peer review (ie, peer reviewers blinded to author names and affiliation) and interventions to increase the speed of the peer-review process. The authors of the systematic review refrain from providing recommendations regarding the wider implementation of the identified interventions due to concerns about their methodological quality, small sample size and applicability. Other researchers have argued that most of the approaches outlined above fail to compensate for potential biases and point out that any success so far has been limited.[12]

Researchers have argued that limited success of interventions might be due to the underlying nature of peer review, which has been described as an intellectual process that encompasses objective and subjective elements.[13] Editors and peer reviewers bring a diverse mix of skills, preferences and intellectual idiosyncrasies to the task.[14] At times, these may result in subjective judgements of manuscript quality. Peer review has also been described as an 'inherently human phenomenon' that is underpinned by social dimensions.[15 16] A qualitative study of the social and subjective dimensions of manuscript peer review in biomedical publishing concluded that biomedical manuscript review '*is a highly social and subjective process driven by communal as well as scientific goals, and influenced by reviewers' and editors' sense of their own authority, power and moral responsibility, as well as by unavoidable prejudice and intuition*'.[17]

Our broader research framework aims to generate an understanding of the communication practices within the editorial and manuscript peer-review process in biomedical research. Disentangling current communication practices for a range of biomedical journals, and outlining the specific roles and tasks of the main actors might be a first step towards establishing the design of interventions that counterbalance social influences on the peer-review process. In this study, we aim to identify and characterise the roles and tasks of the different actors in the process of peer review from the perspective of journal editors.

Our specific objectives are:
1. To examine biomedical editors' experiences of their interactions with peer reviewers and authors.

2. To characterise journal editors' perspectives, expectations, understandings and perceptions regarding the roles and tasks of peer reviewers.

## METHODS AND ANALYSIS
### Qualitative approach and research paradigm
Given its underlying social and subjective dimensions,[17 18] the need for more qualitative research into the peer-review process within the biomedical field has been recognised for some time.[1] However, to date, most such research has been overwhelmingly quantitative in nature.[19]

Drawing on a pragmatist epistemological position that the aim of inquiry cannot be independent from human experience,[20] we considered a qualitative approach to be best suited to answer our research question. The expectations, understandings, perceptions and thoughts of journal editors are largely intangible aspects that cannot be unpacked using predefined categories or viewed independently from the purposes of the peer-review process itself.

The use of qualitative interviews will enable participants to speak freely and at length about such aspects, thus providing rich data embedded in personal experiences and practices.

Data will be analysed using thematic analysis (TA) as described by Braun and Clarke primarily because of the methods flexibility and epistemological assumptions that are compatible with a pragmatist approach.[21]

Study participants will be offered the possibility of conducting the interview either face to face or by phone/conferencing system, according to personal preference and availability. This will also enable the interviewing of people in geographically distant locations.

### Study sample and recruitment
We will use maximum variation purposive sampling to recruit a heterogeneous study sample of biomedical journal editors, allowing us to select editors with different characteristics that we anticipate may influence their perspectives. This sampling method enables conceptual exploration using the characteristics of individuals and journals as the basis of selection in order to reflect the diversity and breadth of the sample population, rather than achieving population representativeness.[22]

Participants will be recruited through different sources. The study recruitment pathway is shown in table 1.

Initially, interviewees will be drawn from a professional network of contacts (members of the Methods in Research on Research (MiRoR) project[23]) who are journal editors.

| Table 1 Study recruitment pathways | |
| --- | --- |
| **Source of participants** | **Sampling** |
| Existing professional networks | Purposive/snowballing |
| Two research publishers | Purposive/snowballing |
| International Congress on Peer Review and Scientific Publication | Purposive/snowballing |

**Table 2** Sample characteristics

| Criteria | Characteristics |
|---|---|
| Demographic characteristics | ► Gender<br>► Editorial experience<br>► Commitment (full-time, part-time)<br>► Editors geographical location |
| Journal characteristics | ► Journal specialty (eg, Clinical, Public Health)<br>► Impact factor (journals with or without impact factor)<br>► Peer-review practices (closed peer review, open peer review, postpublication peer review)<br>► Publisher (medical publishing companies, independent publisher/university)<br>► Open access, paywall<br>► Size (editorial team) |

Four editors will be interviewed for piloting purposes and requested to recommend additional journal editors whom the lead researcher (KG) can interview.

The research publishers BMC (part of Springer Nature) and BMJ are partners of the MiRoR project and this partnership will be used to recruit interviewees. Editors in chief operating within the BMJ Publishing Group will be contacted by the lead researcher via email, provided with study details and asked to either participate themselves or recommend suitable journal editors who can be contacted instead. One follow-up email will be sent after 2 weeks to non-respondents.

Due to a different standard operating procedure, a different recruitment strategy will be employed at BMC. The publishers' communication manager will communicate with all editors via internal mail, introduce the lead researcher and the research and encourage them to respond if interested in participating.

Concurrently, the conference participation lists from the Eighth International Congress on Peer Review and Scientific Publication[24] will be reviewed and potential interviewees will be contacted via email by the lead researcher. One follow-up email will be sent to non-respondents after two weeks.

Following the maximum variation sampling strategy, journal editors who agree to be interviewed will be categorised using the characteristics presented in table 2, some of which have been shown to influence the peer-review process (e.g, gender).[25]

This step will help to determine the characteristics that are under-represented and inform the sampling strategy for identification of further participants in such a way as to maximise the diversity of interviewees.

Lastly, the journal editor identification process will be supplemented through snowball sampling.[26] At the end of each interview, interviewees will be asked to recommend other editors whose experiences might be relevant to the study and who would potentially be interested in contributing to this study. These steps are expected to lead to recommendations that optimise sample variation.

### Saturation

Saturation is a core guiding principle to determine sample sizes in qualitative research, yet few qualitative studies report in detail on the parameters that influenced saturation in their studies.[27] In this study, we will adopt the seven parameters outlined by Hennink et al that influence saturation[28] to establish our sample size determinants and demonstrate the grounds on which saturation will be assessed and achieved, thereby justifying the final sample size. The parameters of saturation and sample size for our study are outlined in table 3. According to Hennink et al, the sample size is determined by the combined influence of all parameters rather than any single parameter alone. In our case, some parameters indicate a smaller sample for saturation and others suggest a larger sample, suggesting the need for an intermediate sample size.

The first parameter is the *purpose* of the study, which in this case is to capture themes from the data using the TA method. The second parameter is *population*. For the purposes of our study, we want do grasp as wide variety

**Table 3** Parameters of saturation and determinants of sample size for our study

| Parameters | Sample size determinant for each parameter | Determinant definition |
|---|---|---|
| Purpose | Capture themes | The thematic analysis method will be used to identify themes and patterns of meanings across the dataset in relation to the research question |
| Population | Heterogeneous | Journal editors with different characteristics (ie, demographic characteristics, journal discipline and characteristics) |
| Sampling strategy | Iterative sampling | Iterative sampling using established networks; enlarged through snowballing |
| Data quality | Thick data | Experiences and opinions will be captured with the aim to provide deep and rich insights |
| Type of codes | Conceptual codes | Explicit and subtle |
| Codebook | Emerging codebook | Emerging codebook existing of inductive and diductive codes updated after every interview |
| Saturation goal and focus | Data saturation | Referring to saturation as the point where no new codes are identified from the data |

of biomedical editors as possible and will thus obtain a heterogeneous sample. This parameter will be satisfied by interviewing journal editors with different characteristics (ie, demographic characteristics, journal specialty and journal characteristics). Our *data collection strategy* will be iterative, involving continual data collection until a sample covering wide variety of experiences and viewpoints has been achieved. We aim to collect *thick data* in order to provide deep and rich insights and capture explicit and concrete codes as well as conceptual codes that capture subtle issues. Our codebook will be *emerging* including a broad range of codes, including explicit, subtle and conceptual codes.

Lastly, the *saturation goal and focus* of our study is to achieve data saturation, that is, the point where no new issues or themes are identified from the data.[28]

Although the process of reaching saturation cannot be meaningfully quantified in advance and involves an iterative approach until saturation is obtained, we used a recently developed quantitative method to offer an initial estimate of expected number of participants in our study. Following the approach suggested by Fugard and Potts[29] of estimating sample size required to achieve code saturation for studies that use TA, we calculated that we would need a sample size of at least 38 participants to detect, with 90% power, two instances of a theme with 10% prevalence. Online supplementary appendix 1 shows the details of the calculation. This is in line with our previously hypothesised number of participants. Therefore, while our core approach to data collection strategy will be iterative, involving continual data collection until saturation is reached, we anticipate around 40 participants to be sufficient to provide us with meaningful information to answer our research questions, in line with similar studies.[17]

### Inclusion criteria and recruitment process

Study participants will consist of journal editors of biomedical journals, referring to individuals who are currently involved in the communication process between authors and peer reviewers and/or who are in a position to decide about the fate of manuscripts. They might also, but not necessarily, contribute to the determination of journal content and policy.

Journal editors will be contacted between October 2017 and February 2018. They will be sent an invitation email and information sheet by the lead author (KG), followed by a phone call to determine if they are interested in participating in the study.

### Interview guide

A preliminary topic guide for the semistructured interviews (see table 4) has been developed, informed by the outcomes of a previously conducted scoping review of the literature.[30] The topic guide was piloted on four editors to assess usefulness and meaningfulness of the questions, the ease of administration, language and length, and to

refine the topic guide. It is likely that the topic guide will be refined further after conducting more interviews.

### Data collection and recording

All interviews will be conducted by the lead researcher (KG) either face to face or by phone or online call (eg, Skype or conferencing system), according to the circumstances and preferences of the interviewees.

With the permission of the participants, interviews will be audio recorded and notes will be taken.

Interviewees will be asked if they could be contacted again if further clarification is needed.

Based on the pilot interviews, it is anticipated that interviews will take around 30 min to complete.

### Data analysis

Data will be analysed using Braun and Clarke's six phase TA described as 'a method for identifying, analysing and reporting patterns (themes) within data'.[21] This analytical framework assumes that truth can be accessed through language, but that accounts and experiences are socially mediated.[31]

It is not bound to any pre-existing theoretical framework, therefore, it offers relative theoretical independence and compatibility with various approaches which is compatible with pragmatist position that we subscribe to.[32] TA has also been described as a more accessible form of analysis compared with other approaches that requires less detailed theoretical and technical knowledge, and is therefore particularly suitable for the lead researcher (KG) of this study who is at an early stage of her qualitative research experience.[21] The lead researcher (KG) will conduct all interviews, which will be transcribed verbatim.

Data analysis will take place concurrently with data collection in an iterative cycle. This serves two purposes: first, it will help to further refine the topic guide and allow the interviewer (KG) to reflect with the senior investigator (DH) on her own interviewing technique and style for subsequent interviews. Second, it will help the researchers to determine when saturation occurs.

The six phases of TA analysis consist of: familiarising with the data, generating initial codes, searching for themes, reviewing themes, defining and naming themes, producing the report.

The first phase will start by familiarising with the data—rereading each transcript at least twice and noting down initial ideas.

In the second phase, initial codes will be generated from a subset of interviews using both, deductive codes from topics in the interview guide and inductive content-driven codes. The codes will be developed line by line from the interview content, focusing on the identification of both semantic (ie, reflecting the explicit content) and latent (ie, reflect the implicit content) features.[21] In order to ensure consistency and credibility, a code manual/codebook will be developed by both researchers (KG and DH). These codes will be then applied to subsequent interviews with sensitivity to the possibility of new

**Table 4** Topic guide for semistructured interviews

| Key area of investigation | Topics | Questions and prompts |
|---|---|---|
| Background information | ▶ Explore personal background<br>▶ Level of experience<br>▶ Own roles and tasks as an editor | ▶ Tell me about your journal and the job you have.<br>▶ How long have you been in this position?<br>Prompt: percentage of time devoted to editorial duties (eg, part time, full time)<br>▶ What are your current responsibilities?<br>▶ Did you hold any other editorial position before your current position? If yes, what were your responsibilities then? |
| Journal set-up | ▶ Explore journal set-up | ▶ Tell me about your journal—how does it work?<br>Prompt: availability of editorial support staff<br>▶ How does the peer-review process work in your journal?<br>▶ What do you do within the process? |
| Opinion on peer-reviewers role and tasks | ▶ Roles and tasks of peer reviewers<br>▶ Expectations | ▶ What do you expect from peer reviewers in terms of their roles and tasks?<br>▶ How do you let your reviewers know what you expect from them?<br>Prompt: on whatever has not been mentioned<br>▶ Attitudes and beliefs (about role and tasks)<br>▶ Organisational expectations (about role and tasks)<br>▶ Can you tell me about a specific situation when you were not satisfied with a review or with a peer reviewer?<br>▶ What did you do in that a situation?<br>Prompt: looks for factors other than being late with a review,or not doing a review once they have accepted it<br>▶ Can you tell me about a situation when you were exceptionally satisfied with a review or with a peer reviewer?<br>▶ Were there situations (in regard to the roles and task of reviewers) when you disagreed with the other editors you work with? What about? What happened?<br>▶ What about other journals, do roles and tasks differ among journals in your field?<br>Prompt: If yes (ie, differences exist), then:<br>▶ How does this affect the process?<br>▶ How does it affect your communication?<br>▶ How do you negotiate those differences? Does it matter? |
| Communication between editors, peer reviewers and authors | ▶ Communication between the three parties<br>▶ Potential conflicts<br>▶ Power | ▶ Can you describe your experience of the communication process between editors, authors and peer reviews?<br>▶ How do you communicate with authors and peer reviewers?<br>▶ Can you give me some specific examples of situations where this communication is challenging?<br>Prompt:<br>▶ What are potential conflicts?<br>▶ When do disagreements arise?<br>▶ What happens if there is disagreement between peer reviewers? |
| Conclusion | ▶ Snowballing<br>▶ Documents<br>▶ Final comments | ▶ Is there anybody else whom you think I should speak to?<br>▶ Any articles/documents I can access/should look at?<br>▶ Any final comments? Is there anything else that you think is important to mention? |

emerging codes that will be added to the code manual and applied to the entire dataset in an iterative manner. The qualitative data analysis software NVivo V.11 will be used to aid data management and analysis (ie, indexing of coding and transcripts).

In the third phase, the codes will be clustered into potential themes to give an indication of their prevalence for the assessment of (code or meaning) saturation, and into a preliminary thematic map displaying the main themes.

The fourth phase will consist of reviewing themes and will be divided into two stages: the reviewing and refining of the data at the level of the coded data extracts, and subsequently at the level of the entire data set. These two

stages will lead to the generation of a thematic map of the analysis.

The aim of the next phase will be to definitively define the scope and content of each relevant theme and precisely name them. This will involve debriefing between the study team. Debriefing with an outside expert (on peer review in biomedical journals) as suggested by King[33] will be conducted to ensure that themes are sufficiently clear to someone outside of the immediate research team.

After the establishment of the final themes, the last phase will consist of writing up the study findings as a journal article. Direct quotes will be used to illustrate specific points of interpretation and the extraction of themes. All themes and subthemes will be presented in the result section and discussed in the light of existing literature.

### Securing study quality

The most widely used criteria for evaluating qualitative analysis are those developed by Lincoln and Guba,[34] who introduced the concept of 'trustworthiness' to parallel the conventional quantitative assessment criteria of validity and reliability. Trustworthiness is determined by applying the concepts of credibility, transferability, dependability and conformability to qualitative research. Credibility corresponds to the concept of validity, whereby researchers seek to ensure that a study measures what it is actually intended to measure. Transferability corresponds to external validity, or the extent to which the research can be transferred to other contexts. Dependability corresponds with reliability, or whether the research process is methodologically consistent and correct, whether the research questions are clear and logically connected to the research purpose and design, and whether findings are consistent and repeatable. Confirmability is concerned with establishing that the researcher's interpretations and findings are clearly derived from the data, requiring the researcher to demonstrate how conclusions and interpretations have been reached.[35]

In order to establish trustworthiness in this research, the step-by-step approach proposed by Nowell *et al*—which provides a detailed description of how to conduct a trustworthy TA—will be followed.[36] These authors use the criteria by Lincoln and Guba and show how these can be achieved throughout the six phases of TA.

We will use reporting guidelines for reporting qualitative research to provide detailed reporting of methods used.[37]

### Patient and public involvement

There will be no patient or public involvement in this research project.

### DISCUSSION

This research has multiple potential uses. As a standalone research piece, it will generate context-based information from journal editors' perspectives that will help to provide insight into the communication patterns within biomedical journals, including differences and similarities across biomedical journals. It is also embedded within a larger project that will inform the analysis of peer-reviewer reports.

The study findings can further be used to inform biomedical journal policies and develop training courses for peer reviewers and journal editors.

### Ethics and dissemination

Interviewees will receive an information sheet about the research and a consent form before the interview. The information letter includes details on the maintenance of anonymity and confidentiality throughout the research process. Prior to the interview, information from the information sheet and consent form will be reiterated verbally, and interviewees will be asked to consent to participation and recording of their interviews. Participants will be able to choose not to be directly quoted in any publications resulting from the study.

Findings will be disseminated through a publication in a peer-reviewed journal and presentations at academic conferences and other meetings.

**Acknowledgements** The authors would like to thank Professor Erik Cobo and Dr Daniel Cauchi for providing advice during the writing of this protocol and Sara Schroter (BMJ) and Elizabeth Moylan (BMC) for providing guidance and help on the recruitment strategy of interviewees.

**Contributors** All authors have made substantive intellectual contributions to the development of this protocol. KG conceptualised the study approach and led the writing of the manuscript. DH led the supervision of the manuscript preparation. DH was involved in developing the study questions and design and provided detailed comments on earlier drafts.

**Funding** This project was supported by the European Union's Horizon 2020 research and innovation programme under the Marie Sklodowska-Curie grant agreement No 676207.

**Disclaimer** The funders had no role in the study design, data collection and analysis, decision to publish or preparation of the manuscripts.

**Competing interests** At the time of the submission of this protocol KG conducted a secondment at the BMJ.

**Patient consent** Not required.

**Ethics approval** This project has been evaluated and approved by the University of Split, Medical School Ethics Committee (2181-198-03-04-17-0029) in May 2017.

**Provenance and peer review** Not commissioned; externally peer reviewed.

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
