## [Reviewer comments · BMJ Open]

ARTICLE DETAILS

TITLE (PROVISIONAL)	The editors' perspectives on the peer review process in biomedical journals: Protocol for a qualitative study
AUTHORS	Glonti, Ketevan; Hren, Darko

VERSION 1 – REVIEW

REVIEWER	Victoria S. S. Wong University of Hawaii, Honolulu, HI, USA.
REVIEW RETURNED	30-Nov-2017

GENERAL COMMENTS	Although this protocol's goal is framed as generating further understanding of "communication practices" between authors, editors, and peer reviewers, I'm not sure that's the question this study will answer. In my experience, communication in the peer review process consists of the editor sending out form e-mail requests to potential peer reviewers, peer reviewers filling out the journal's standard electronic peer review form and providing comments, authors receiving peer reviewer and editor comments in an e-mail, etc. – there's really very little personalized communication between the editor and the other parties. To get at the actual content of editor-reviewer communications, looking at peer review forms would be more fruitful (e.g., Hirst et al. PLoS One. 2012;7:e35621). The true research question here seems to be at the bottom of page 3, within the Methods section, about the "expectations, understandings, perceptions, and thoughts of journal editors." Journal editors' expectations and perceptions of peer reviewers and authors are no less important than "communication practices" in defining "social influences" on the peer review process, but I would recommend the Introduction frame the background differently. Consider citing this study about what editors say at manuscript meetings including mentions of peer reviewers (Dickersin et al. BMC Med Res Methodol. 2007;7:44.) or this study on journal editor expectations of peer reviewers (Chauvin et al. BMC Med. 2015;13:158.). My knowledge of qualitative research methodology is limited, so I am unable to comment on whether use of the Braun and Clarke analysis method, study recruitment methods, the "saturation" principle, the Fugard and Potts approach, and the Nowell et al. approach are appropriate in this study context. The manuscript mentions "Nowell et al." twice. I think the authors mean "Nowell et al."? Including the response rate will be important since there may be a significant difference between characteristics of editors who are
---

	responders and non-responders – how will this be tracked given the “snowballing” method of trying to recruit study participants? In Table 3, under journal characteristics, consider inclusion of total journal publications and citation counts. Total citation count may be a better measure of overall impact than impact factor. The inclusion criteria for study participants is not quite right: “journal editors... who are currently involved in the communication process between authors and peer reviewers and therefore in a position to decide about the fate of manuscripts.” Sometimes there is a managing editor who is responsible for most or all of the communication with authors and peer reviewers, yet is not the handling editor for the manuscript. (Reference: https://www.editorialmanager.com/robohelp/6.0/Assigning_submissions_to_Editors___Editor_Chains.htm) Due to the inconsistent definitions and roles of different editor types across journals (e.g., deputy editors, associate editors, senior editors, executive editors, section editors, assistant editors, consulting editors, etc.), asking interviewees for their formal position title and then asking what they actually do would be an interesting exercise. Will any interview questions be sent to interviewees beforehand so they can prepare? Will editors-in-chief be treated any differently in this study? They have quite a different role in the entire process. Will they be excluded? If not, will they be asked the same questions? Other questions to consider asking interviewees may include what percentage of their time they devote to their editorial duties (e.g., part time, full time, or the number of manuscripts handled per year), their process for finding peer reviewers, availability of editorial support staff, journal expectations related to peer review turnaround time and number of peer reviewers per manuscript, and whether they consider the strengths and weaknesses of their chosen peer reviewers. With regard to editor perception of peer reviewer roles and tasks, consider adding specific prompts such as expectations about review of content, methodology, statistics, spelling and grammar, etc. It is known that peer reviewers have a very strong influence on the fate of manuscripts despite very poor agreement between reviewers (Rothwell et al. Brain. 2000;123:1964). It would be interesting to see how editors interpret this finding. The authors mention on page 3, objective #3, that one study goal is to explore journal editors’ views on peer review in relation to the fate of individual manuscripts, yet this is not part of the interview questions. In the entire manuscript, there is a heavy emphasis on peer reviewers, so any mention of editor communication with and perceptions of authors seems like an afterthought. Consider either getting rid of the part about authors, or fleshing out that part of the interview further. What are editors’ “expectations, understandings, perceptions, and thoughts” of authors?
--	--

	Limitations are not well defined but presently include typical limitations of qualitative research (including dependability of the information provided by interviewees), use of purposive/snowballing methods of study recruitment resulting in bias, as well as small sample size. Even though a power calculation was performed to determine sample size, one can imagine that speaking with only 40 journal editors is nowhere near a fair representation of all biomedical journal editors. Note: I have no conflicts of interest, personal or professional, with regard to all above article citations.
--	---

REVIEWER	John D. Bowman Texas A&M Health Science Center Irma Lerma Rangel College of Pharmacy, Pharmacy Practice
REVIEW RETURNED	04-Dec-2017

GENERAL COMMENTS	I have no comments. I think the proposal is worthwhile but the methodology is beyond my scope.
--

REVIEWER	Amelia Gibson University of North Carolina at Chapel Hill
REVIEW RETURNED	07-Feb-2018

GENERAL COMMENTS	Thank you for the opportunity to review this protocol, "The editors' perspectives on communication practices within the manuscript review process in biomedical journals: Protocol for a qualitative study." It is a timely and valuable contribution to the methodological literature, and the resulting study will address a topic that is of continued importance to biomedical researchers and reviewers alike. The paper is very informative, and will be instructive for qualitative researchers looking to design representative samples. There are a few comments in the attached document, mainly related to clarifying details of the protocol. It would also help for the author to clarify the difference between this version of thematic analysis, with theoretical sampling, etc. and grounded theory research. The differentiation is mentioned, but not explained in any detail. Again, this is a valuable methodological paper, and will be immensely useful. - The reviewer provided a PDF file with additional comments. Please contact the publisher for full details.
---

REVIEWER	Dr Shazia Jamshed Pharmacy Practice Department, Kulliyah of Pharmacy, International Islamic University Malaysia, Kuantan, Pahang, 25200, Malaysia.
REVIEW RETURNED	27-Mar-2018

GENERAL COMMENTS	Dear Author, First of all it is great pleasure to read your study protocol which is detail oriented and comprehensively written. I just want to highlight that for the evaluation of interpretive research a criteria needs to be followed. It is suggested that you may include Guba and Lincoln criteria for the evaluation of interpretive research. It will add robust look to the study protocol.
--

VERSION 1 – AUTHOR RESPONSE

Reviewers' Comments to authors and replies

Reviewer: 1

Although this protocol's goal is framed as generating further understanding of “communication practices” between authors, editors, and peer reviewers, I'm not sure that's the question this study will answer. In my experience, communication in the peer review process consists of the editor sending out form e-mail requests to potential peer reviewers, peer reviewers filling out the journal's standard electronic peer review form and providing comments, authors receiving peer reviewer and editor comments in an e-mail, etc. – there's really very little personalized communication between the editor and the other parties. To get at the actual content of editor-reviewer communications, looking at peer review forms would be more fruitful (e.g., Hirst et al. PLoS One. 2012;7:e35621). The true research question here seems to be at the bottom of page 3, within the Methods section, about the “expectations, understandings, perceptions, and thoughts of journal editors.” Journal editors' expectations and perceptions of peer reviewers and authors are no less important than “communication practices” in defining “social influences” on the peer review process, but I would recommend the Introduction frame the background differently. Consider citing this study about what editors say at manuscript meetings including mentions of peer reviewers (Dickersin et al. BMC Med Res Methodol. 2007;7:44.) or this study on journal editor expectations of peer reviewers (Chauvin et al. BMC Med. 2015;13:158.).

- Thank you for this very useful, elaborate comment. In this study we explore the expectations, understandings, perceptions, and thoughts of editors regarding peer review, with the aim of disentangling current communication practices and outlining the specific roles and tasks of peer reviewers. We believe that social influences on the peer review process are manifested through certain communication practices – we therefore think it is appropriate to frame the introduction to the study around the broader picture of communication practices. However, we agree that the emphasis is on the peer reviewer aspect, and that this should be explained more clearly. Based on your suggestions we have now changed:
 - 1) The title of the manuscript to: “The editors' perspectives on the peer review process in biomedical journals: Protocol for a qualitative study”
 - 2) Reframed the introduction in accordance with your suggestions, using the last reference that you kindly provided (page 3) to:

“A study that aimed to identify all tasks that are expected of peer reviewers by journal editors when evaluating a manuscript reporting a randomized controlled trial (RCT) found that the most important tasks in peer review, as perceived by peer reviewers evaluating RCTs, were not congruent with the tasks most often requested by journal editors in their guidelines to reviewers (4) These differences illustrate the need to clarify the roles and tasks of peer reviewers.”
 - 3) Changed the specific objectives (page 3) to:
 - 1) To examine biomedical editors' experiences of their interactions with peer reviewers and authors
 - 2) To characterize journal editors' perspectives, expectations, understandings and perceptions regarding the roles and tasks of peer reviewers

My knowledge of qualitative research methodology is limited, so I am unable to comment on whether use of the Braun and Clarke analysis method, study recruitment methods, the “saturation” principle, the Fugard and Potts approach, and the Nowell et al. approach are appropriate in this study context.

- Thank you for acknowledging this, we appreciate your candour. We have provided a number of reasons (see page 9 and 10) why we think that these approaches are best suited for our study.

The manuscript mentions “Nowell at al.” twice. I think the authors mean “Nowell et al.”?

- We have now corrected these typos.

Including the response rate will be important since there may be a significant difference between characteristics of editors who are responders and non-responders – how will this be tracked given the “snowballing” method of trying to recruit study participants?

- Thanks for raising this important point. In contrast to quantitative studies involving larger numbers of participants, response rate in qualitative studies is not an essential or relevant component. Our qualitative research aims for a deep understanding of how a diverse group of journal editors derive meaning from the peer review process at their journal. Therefore, the characteristics of non-responders is not relevant to our chosen approach. We are following the available reporting guidelines for qualitative research where the aspect of responders vs non-responders – in contrast to reporting guidelines for quantitative research – does not appear:
 - Tong A, Sainsbury P, Craig J. Consolidated criteria for reporting qualitative research (COREQ): a 32-item checklist for interviews and focus groups. *International journal for quality in health care*. 2007 Dec 1;19(6):349-57.
 - O’Brien BC, Harris IB, Beckman TJ, Reed DA, Cook DA. Standards for reporting qualitative research: a synthesis of recommendations. *Academic Medicine*. 2014 Sep 1;89(9):1245-51.

In Table 3, under journal characteristics, consider inclusion of total journal publications and citation counts. Total citation count may be a better measure of overall impact than impact factor.

- Thank you for this suggestion. In drawing up this table, we were concerned about inadvertently providing information that would lead to identification of journals and study participants, particularly those publishing in small, very specialized areas and/or different countries. In order to avoid this, journal characteristics in the table are purposefully vague – restricted to availability or no availability of an impact factor. Since we targeted very different fields of biomedicine and journals of different sizes, we feel that the additional value of presenting total number of journal publications and citation counts is limited.

The inclusion criteria for study participants is not quite right: “journal editors... who are currently involved in the communication process between authors and peer reviewers and therefore in a position to decide about the fate of manuscripts.” Sometimes there is a managing editor who is responsible for most or all of the communication with authors and peer reviewers, yet is not the handling editor for the manuscript. (Reference: https://www.editorialmanager.com/robohelp/6.0/Assigning_submissions_to_Editors_Editor_Chains.htm). Due to the inconsistent definitions and roles of different editor types across journals (e.g., deputy editors, associate editors, senior editors, executive editors, section editors, assistant editors, consulting editors, etc.), asking interviewees for their formal position title and then asking what they actually do would be an interesting exercise.

- Thank you for your suggestion. We have now updated our inclusion criteria (page 7) to demonstrate eligibility of a broad range of participants i.e. that study participants have to be

involved in the communication process between authors and peer reviewers and/or be in the position to decide on the fate of the manuscript:

“Study participants will consist of journal editors of biomedical journals, referring to individuals who are currently involved in the communication process between authors and peer reviewers and/or be in a position to decide about the fate of manuscripts. They might also, but not necessarily, contribute to the determination of journal content and policy.”

Your following point about inconsistent definitions of job titles is interesting, and we fully agree that there are inconsistencies between definitions and roles. However, the focus of our study is not to highlight these but talk to interviewees who meet the inclusion criteria. Nevertheless, at the beginning of each interview every participant will be asked about their position and asked to describe his/her role and responsibilities in the peer review process in detail (please refer to first questions of the topic guide). Thus, displaying the different roles and tasks of the interviewees.

Will any interview questions be sent to interviewees beforehand so they can prepare?

- Interview questions will not be sent to interviewees beforehand. This procedure is standard in qualitative research. Furthermore, we are using a semi-structured interview approach (see page 8), i.e. it is a flexible allowing the interviewer to react/adjust to the responses of the interviewee and ask further clarification questions.

Will editors-in-chief be treated any differently in this study? They have quite a different role in the entire process. Will they be excluded? If not, will they be asked the same questions?

- According to our inclusion criteria editors are eligible if they are involved in the communication between authors and peer reviewers or if they take a decisions on the fate of the manuscript (which in turn is informed by peer reviewers' reports). As mentioned in your previous comment, the roles of editors varies across journals. This is also the case for editors-in-chief who perform different tasks according to journal set-up, size etc. Therefore editors-in-chief will not be treated differently to other editors, and will be considered as potential study participants.

Other questions to consider asking interviewees may include what percentage of their time they devote to their editorial duties (e.g., part time, full time, or the number of manuscripts handled per year), their process for finding peer reviewers, availability of editorial support staff, journal expectations related to peer review turnaround time and number of peer reviewers per manuscript, and whether they consider the strengths and weaknesses of their chosen peer reviewers. With regard to editor perception of peer reviewer roles and tasks, consider adding specific prompts such as expectations about review of content, methodology, statistics, spelling and grammar, etc.

- Thank you for these suggestions. The suggested prompts have now been added in the revised topic guide.

It is known that peer reviewers have a very strong influence on the fate of manuscripts despite very poor agreement between reviewers (Rothwell et al. *Brain*. 2000;123:1964). It would be interesting to see how editors interpret this finding. The authors mention on page 3, objective #3, that one study goal is to explore journal editors' views on peer review in relation to the fate of individual manuscripts, yet this is not part of the interview questions.

- Thank you for raising this very relevant point. We considered this aspect to be already included in the topic guide section “Communication between editors, peer reviewers and authors” under

the question: “Can you give me some specific examples of situations where this communication is challenging?”. However, we acknowledge that it could have been clearer and have now introduced a specific prompt into the topic guide in relation to disagreement or agreement between reviewers.

Based on your first comment we have reframed our study objectives (page 3). Study objective 3 was not meant to look at disagreements between peer reviewers. We have now removed it to avoid misunderstanding.

In the entire manuscript, there is a heavy emphasis on peer reviewers, so any mention of editor communication with and perceptions of authors seems like an afterthought. Consider either getting rid of the part about authors, or fleshing out that part of the interview further. What are editors’ “expectations, understandings, perceptions, and thoughts” of authors?

- Our research question is specifically focused on the editors’ understanding of the roles and tasks of peer reviewers, not of authors, hence the existing emphasis. However, we are also interested in the communication between the three parties, so we included three questions that relate to authors. Given that journal editors are typically authors and peer reviewers as well, we think that this aspect should not be completely excluded because it will add to the richness of the data, but it also need not be a core area for exploration.

Limitations are not well defined but presently include typical limitations of qualitative research (including dependability of the information provided by interviewees), use of purposive/snowballing methods of study recruitment resulting in bias, as well as small sample size. Even though a power calculation was performed to determine sample size, one can imagine that speaking with only 40 journal editors is nowhere near a fair representation of all biomedical journal editors.

- Thank you raising these important points. In the ‘Strengths and Limitations of the Study’ section on page 2, we acknowledge the fact that qualitative research findings cannot directly be extrapolated to all biomedical journals and other scientific fields. Qualitative study approaches cannot and should not claim to provide a fair representation of a population; rather, we hope to offer a deeper insight into the experiences of a variety of editors.
We believe that the potential limitations you described are more applicable to quantitative study design.
With regards to purposive sampling and snowballing, these are legitimate approaches to recruit study participants. Purposeful sampling is widely used in qualitative research for the identification and selection of potentially “information-rich” cases related to the phenomenon of interest and facilitates recruitment of participants.
With regards to small sample size – based on the outlined parameters of saturation (by Hennink et al., see page 6) our aim is to conduct around 40 interviews. This is considered to be an intermediate to big sample in qualitative research.

Reviewer: 2

I have no comments. I think the proposal is worthwhile but the methodology is beyond my scope.

- Thank you for your agreement that the proposal is worthwhile, and we appreciate your candour regarding your knowledge of qualitative research methodology.

Reviewer: 3

Thank you for the opportunity to review this protocol, "The editors' perspectives on communication practices within the manuscript review process in biomedical journals: Protocol for a qualitative study." It is a timely and valuable contribution to the methodological literature, and the resulting study will address a topic that is of continued importance to biomedical researchers and reviewers alike. The paper is very informative, and will be instructive for qualitative researchers looking to design representative samples. There are a few comments in the attached document, mainly related to clarifying details of the protocol.

- Thank you for your agreement that the proposed study is worthwhile and the protocol useful. Below we have extracted your comments and created a point-by-point response.

...it is unclear which citation (9? 10?) is the citation to the recent systematic review, and which are citations to definitions or specific examples. It might be best to place the citation directly following the words "biomedical publications."

- The examples provided are from the systematic review. We have now placed the citation directly following the words 'biomedical publications'. See page 2:

"A recent systematic review evaluating the impact of interventions to improve the quality of peer review for biomedical publications (10) identified 25 strategies that have been implemented, including training interventions; use of checklists (such as Consort (9)); addition of specific experts (i.e. statistical peer reviewers); the introduction of open peer review (i.e., peer reviewers informed that their identity would be revealed) or blinded peer review (i.e., peer reviewers blinded to author names and affiliation); and interventions to increase the speed of the peer review process."

It is unclear whether this refers to the authors of this paper or the authors of the systematic review that was just discussed.

- This refers to the paper of the systematic review that was discussed in the previous sentence. We have now added this clarification into the sentence that you commented on (page 2): "The authors of the systematic review refrain from providing recommendations regarding the wider implementation of the identified interventions due to their low methodological quality."

I am unclear as to what this means. Earlier discussion warning me of bias against certain types of research (lines 38-40) has primed me to be cautious about statements like this. I would offer either some clarification, a direct quote, or a citation to support this label (low methodological quality).

- Thank you for pointing this out. The sentences you are referring to (lines 38-40) refer to studies that have identified bias within the peer review process, whereas the sentence that you are commenting on is in a new paragraph and is about a systematic review evaluating the impact of interventions to improve the quality of peer review for biomedical publications (see lines 46-47). The authors of the systematic review assess the risk of bias of the included interventions (RCTs) by following domains of the commonly used Cochrane Collaboration Risk of Bias tool. They comment that "The limited number of RCTs identified, their small sample sizes, their methodological quality, and their applicability limit the interpretation of our results.". In order to clarify that we are referring to the systematic review authors' cautious comments, the sentence has been changed to:
"The authors of the systematic review refrain from providing recommendations regarding the wider implementation of the identified interventions due to concerns about their low methodological quality, small sample size and applicability."

How many will be in this initial pilot sample?

- This is specified later on in the text on page 7. However, we have now also added this information on page 4 where your comment appears:
“Four editors will be interviewed for piloting purposes and requested to recommend additional journal editors whom the lead investigator can interview.”

What is the target sample size? If this is a theoretical sample, it would help to look to Charmaz, etc. on GT methodology. Otherwise, proposing a sample size would help clarify the protocol.

- Target sample size (around 40 interviews) and saturation is described in detail on page 6 and 7. We are not using grounded theory methodology but thematic analysis by Braun and Clarke as described in detail on page 9 and 10.

Are the researchers certain that the organizational structure that necessitates this change in procedure is unique to this organization? If not, they might find a great deal more participation from BMC, and less from other organizations with similar structures (that did not employ this protocol).

- Thank you for raising this valid point. We agree, that we might find more participants from BMC. However, the most important aspect for our study is to recruit a heterogeneous sample of journal editors across different biomedical disciplines and from different journals. BMC has a wide range of journals that differ in size, impact, focus and peer review processes, therefore we expect to recruit a diverse group of editors from this publisher.

Eighth?

- We have now corrected this typo.

This section is particularly well-written and informative/instructive for other qualitative researchers.

- Thank you for this positive feedback.

I am unclear as to what the headers for columns 2 and 3 mean, exactly.

- Thank you for pointing this out. We use the seven parameters that influence saturation outlined by Hennink et al. to identify our sample size determinants and demonstrate the grounds upon which saturation will be assessed and achieved, thereby justifying the final sample size (see page 6). The parameters of saturation and sample size for our study are outlined in Table 4. Column 2 represents the sample size determinants that relate to the parameters in the first column. We have now re-named it to “Sample size determinant for each parameter” to clarify this. Each determinant is further defined through concrete references from our study in column 3 which we now re-named to “Determinant definition”.

These dates have already passed.

- This is due to the time lag in protocol submission and time of peer review.

So, the researchers will code in excel for the first round, and Nvivo for the second? Is there a reason for this?

- An Excel spreadsheet will be used to log all details related to data collection (e.g. interview schedule, field notes) processing (e.g. transcription, coding and analysis) and progress throughout. Thus, Excel is only used for data management purposes while data coding will be performed in NVivo. We have now removed this sentence from the “data analysis” section to avoid potential misunderstanding of the coding approach.

It would also help for the author to clarify the difference between this version of thematic analysis, with theoretical sampling, etc. and grounded theory research. The differentiation is mentioned, but not explained in any detail. Again, this is a valuable methodological paper, and will be immensely useful.

- Thank you for this comment. We believe that a detailed elaboration of the differences between Grounded Theory and Thematic Analysis is beyond the scope of this paper and has already been described elsewhere. Our protocol is not a methodological paper that aims to compare different qualitative research methods. Instead, we elaborate upon our methodological approach in some detail to ensure transparency, quality and reproducibility of the proposed qualitative study. We have outlined in detail why we believe that analysing interviews using the thematic analysis approach by Braun and Clarke is our preferred approach (page 9). We have now removed the reference to grounded theory to avoid any misunderstanding.

Reviewer: 4

First of all it is great pleasure to read your study protocol which is detail oriented and comprehensively written. I just want to highlight that for the evaluation of interpretive research a criteria needs to be followed. It is suggested that you may include Guba and Lincoln criteria for the evaluation of interpretive research. It will add robust look to the study protocol.

- Thank you for raising this important point. We refer to the publication by Nowell et al. that base their approach on the Lincoln and Guba criteria. However, we agree that this is a fundamental principle and should be explained in our protocol as well. We have now included a paragraph on page 10-11:
 “The most widely used criteria for evaluating qualitative analysis are those developed by Lincoln and Guba, who introduced the concept of ‘trustworthiness’ to parallel the conventional quantitative assessment criteria of validity and reliability. Trustworthiness is determined by applying the concepts of credibility, transferability, dependability and conformability to qualitative research. Credibility corresponds to the concept of validity, whereby researchers seek to ensure that a study measures what it is actually intended to measure. Transferability corresponds to external validity, or the extent to which the research can be transferred to other contexts. Dependability corresponds with reliability, or whether the research process is methodologically consistent and correct, whether the research questions are clear and logically connected to the research purpose and design, and whether findings are consistent and repeatable. Confirmability is concerned with establishing that the researcher’s interpretations and findings are clearly derived from the data, requiring the researcher to demonstrate how conclusions and interpretations have been reached.”

VERSION 2 – REVIEW

REVIEWER	Victoria Wong
-----------------	---------------

	The John A. Burns School of Medicine at the University of Hawaii at Manoa; Department of Medicine., USA, Honolulu, HI.
REVIEW RETURNED	29-May-2018

GENERAL COMMENTS	The first line on page 8 needs a space between “bein” (“be in”) but even that doesn’t make grammatical sense in the context of the full sentence. Within Table 5, for the background question of “Did you hold any other positions in the same field before your current position?” does this specifically means editorial positions? No other comments -- my prior review's concerns were addressed adequately by the authors.
--

REVIEWER	Shazia Jamshed IIUM Malaysia
REVIEW RETURNED	17-May-2018

GENERAL COMMENTS	The authors successfully addressed all the comments
---

VERSION 2 – AUTHOR RESPONSE

Reviewer: 1

The first line on page 8 needs a space between “bein” (“be in”) but even that doesn’t make grammatical sense in the context of the full sentence.

- We have now corrected this typo and improved the grammar of the sentence:

“Study participants will consist of journal editors of biomedical journals, referring to individuals who are currently involved in the communication process between authors and peer reviewers and/or who are in a position to decide about the fate of manuscripts.”

Within Table 5, for the background question of “Did you hold any other positions in the same field before your current position?” does this specifically means editorial positions?

- Thank you for picking this up. We have now refined this question to:

“Did you hold any other editorial position before your current position?”

No other comments -- my prior review's concerns were addressed adequately by the authors.

Reviewer: 4

The authors successfully addressed all the comments.